



# Sea-Ice Indicators of Polar Bear Habitat

Harry L. Stern[1] and Kristin L. Laidre[1,2]

5    [1]Polar Science Center, Applied Physics Laboratory, University of Washington
1013 NE 40th Street, Seattle, WA 98105 USA

[2]Greenland Institute of Natural Resources
Box 570, 3900 Nuuk, Greenland

*Correspondence to*: Harry L. Stern (harry@apl.washington.edu)

**Abstract.** Nineteen distinct subpopulations of polar bears (*Ursus maritimus*) are found throughout the Arctic, and in all regions they depend on sea ice as a platform for traveling,

hunting, and breeding. Therefore polar bear phenology – the cycle of biological events – is tied to the timing of sea-ice retreat in spring and advance in fall. We analyzed the dates of sea-ice retreat and advance in all 19 polar bear subpopulation regions from 1979 to 2014, using daily sea-ice concentration data from satellite passive microwave instruments. We define the dates of sea-ice retreat and advance in a region as the dates when the area of sea ice drops below a certain

threshold (retreat) on its way to the summer minimum, or rises above the threshold (advance) on its way to the winter maximum. The threshold is chosen to be halfway between the historical (1979-2014) mean September and mean March sea-ice areas. In all 19 regions there is a trend toward earlier sea-ice retreat and later sea-ice advance. Trends generally range from −3 to −9 days decade$^{-1}$ in spring, and from +3 to +9 days decade$^{-1}$ in fall, with larger trends in the Barents

Sea and central Arctic Basin. The trends are not sensitive to the threshold. We also calculated the number of days per year that the sea-ice area exceeded the threshold (termed ice-covered days), and the average sea-ice concentration from June 1 through October 31. The number of ice-covered days is declining in all regions at the rate of −7 to −19 days decade$^{-1}$, with larger trends in the Barents Sea and central Arctic Basin. The June-October sea-ice concentration is

declining in all regions at rates ranging from −1 to −9 percent decade$^{-1}$. These sea-ice metrics (or indicators of change in marine mammal habitat) were designed to be useful for management agencies. We recommend that the National Climate Assessment include the timing of sea-ice retreat and advance in future reports.

**Keywords**: Arctic sea ice, polar bears, timing of sea-ice retreat and advance





## 1 Introduction

The International Union for Conservation of Nature (IUCN) Polar Bear Specialist Group (PBSG) recognizes 19 subpopulations of polar bears (*Ursus maritimus*) (Obbard et al., 2010) (Fig. 1 and

Table 1). They are found throughout the sea-ice-covered areas of the circumpolar Arctic, especially over the continental shelf and inter-island channels. Polar bears depend on sea ice as a platform for hunting. Sea ice also facilitates seasonal movements, mating, and, in some areas, maternal denning (Wiig et al., 2015). Some polar bears remain on sea ice year-round, but in more southerly areas where the ice melts completely, all bears are forced to spend up to several

months on land, fasting until freeze-up allows them to return to the ice again (e.g., Stirling et al., 1999; Stirling and Parkinson, 2006). The global population size of polar bears is estimated to be about 25,000 (Obbard et al., 2010). Genetic analysis indicates that there is considerable gene flow between some subpopulations, while others are relatively discrete (Paetkau et al., 1999; Peacock et al., 2015).

Multiple approaches have been taken to construct sea-ice metrics for studies of survival and body condition in specific polar bear subpopulations (Table 2). These have generally focused on subpopulation-specific metrics such as the number of ice-free or ice-covered days per year (Obbard et al., 2007; Regehr et al., 2010; Hamilton et al., 2014; Regehr et al., 2015), the dates of spring sea-ice breakup and/or fall sea-ice freeze-up (Stirling and Parkinson, 2006;

Regehr et al., 2007; Lunn et al., 2014; Laidre et al., 2015a; Obbard et al., 2016), or the sea-ice concentration (Rode et al., 2012; Peacock et al., 2012; Peacock et al., 2013). Sea-ice metrics have mainly been selected based on the specific region under study, or developed for single studies or data sets. There is a need to develop standardized circumpolar metrics of polar bear habitat based on the satellite record of sea ice that allow for regional comparisons of habitat

change and for tracking changes into the future.

In this study we used daily sea-ice concentration data to calculate several sea-ice metrics for each of the 19 polar bear subpopulation regions for the period 1979-2014. The metrics are: date of spring sea-ice retreat, date of fall sea-ice advance, average sea-ice concentration from June 1 to October 31, and the number of ice-covered days per year. We calculated each metric

for the total marine area of each region, and for the shallow depths only (less than 300 m).



Shallow depths are more biologically productive and are considered to be better polar bear habitat (Wiig et al., 2015).

Several previous studies have divided the Arctic into distinct regions and calculated the sea-ice area trend in each region (e.g., Stroeve et al., 2012; Perovich and Richter-Menge, 2009; Parkinson and Cavalieri, 2008). While this is a straightforward and useful way to document changes in sea ice, other metrics of sea-ice habitat are more relevant to marine mammals that are behaviorally tied to the annual retreat of sea ice in the spring and advance in the fall. Many ecologically important regions of the Arctic are ice-covered in winter and ice-free in summer, and will probably remain so for a long time into the future. Therefore the dates of sea-ice retreat in spring and advance in fall, and the interval of time between them, are key indicators of climate change for ice-dependent marine mammals.

**2 Data**

As in Laidre et al. (2015a) we used the *Sea Ice Concentrations from Nimbus-7 SMMR and DMSP SSM/I-SSMIS Passive Microwave Data* (Cavalieri et al., 1996, updated yearly) available from the National Snow and Ice Data Center (NSIDC) in Boulder, Co. This product is designed to provide a consistent time series of sea-ice concentrations (the fraction, or percentage, of ocean area covered by sea ice) spanning the coverage of several passive microwave instruments. The sea-ice concentrations are produced using the NASA Team algorithm, and are provided in a polar stereographic projection (true at 70°N) with a nominal grid cell size of $25 \times 25$ km. (Cell size varies slightly with latitude). Temporal coverage is every other day from 26 October 1978 through 9 July 1987, then daily through 31 December 2014.

Concerning the accuracy of the sea-ice concentration data, the product documentation states that it is within ±5% of the actual sea-ice concentration in winter, and ±15% in summer when melt ponds are present on the sea ice (NSIDC, 2015). We note that averaging over many grid cells, as is done here for the 19 regions, greatly reduces the random component of the error, although it would not reduce a bias, if present.

The spatial coverage of the sea-ice concentration data excludes a small circle around the North Pole, due to the satellite orbits. This "pole hole" is entirely surrounded by the Arctic Basin region (AB in Fig. 1 and Table 1). Although the size of the pole hole became smaller in



1987 with the advent of a new satellite and instrument, we use the larger pre-1987 pole hole for consistency of calculations throughout the period 1979-2014. Our Arctic Basin region does not include the pole hole; it surrounds the pole hole.

For bathymetry we used ETOPO1, a 1 arc-minute global relief model of Earth's surface that integrates land topography and ocean bathymetry, built from numerous global and regional data sets (Amante and Eakins, 2009). We averaged the ETOPO1 data over each SSM/I grid cell to get the mean ocean depth for each cell, which we used to distinguish the continental shelf (less than 300 meters depth) from the deeper ocean. Table 1 gives the marine area of the 19

subpopulation regions, as well as the percent of the area shallower than 300 meters and deeper than 300 meters.

## 3 Methods

### 110 3.1 Preliminary data processing

Sea-ice area is defined as (sea-ice concentration) × (grid cell area) summed over cells with sea-ice concentration greater than 15%. For each region, we calculated the daily (or every-other-day prior to 1987) sea-ice area over two sets of grid cells: (1) all cells in the region, and (2) those

cells in which the mean ocean depth is less than 300 meters.

        We next looked for outliers in each time series: excessively large or small values that may be the result of erroneous sea-ice retrievals due to extreme weather events or other errors. Outliers were identified by comparing each value in the time series with a 5-point median-filtered version of the time series. If the difference between the actual value and the median-

filtered value exceeded a certain threshold (15% of the mean March sea-ice area), then the actual value was replaced by the median value. The outlier rate was less than three values per 10,000. This procedure also led to the identification of an anomaly on 14 September 1984 that we reported to NSIDC, and which turned out to be an error in the passive microwave source data (see Product History at http://nsidc.org/data/docs/noaa/g02135_seaice_index/). NSIDC

subsequently re-processed the data for that day.



We next used linear interpolation to fill in the every-other-day gaps up to 9 July 1987. We also used linear interpolation to span a data gap from 3 December 1987 to 13 January 1988. The end result was a complete time series of daily sea-ice area for each region, 1979-2014.

**3.2 Dates of spring sea-ice retreat and fall sea-ice advance**

The date of spring sea-ice retreat is defined here as the date when the sea-ice area drops below a certain threshold on its way to the summer minimum. The date of fall sea-ice advance is defined as the date when the sea-ice area rises above the threshold on its way to the winter maximum. These dates may or may not occur in what is normally considered to be spring or fall; they are meant to mark the transitions between winter and summer sea-ice conditions.

Arctic sea ice typically reaches its maximum area in March and its minimum area in September. Accordingly, we chose the transition threshold for each region as follows. We calculated the mean March sea-ice area over the period 1979-2014, and the mean September sea-ice area over the same period, and then chose the transition threshold to be halfway between these means. This is illustrated for the Baffin Bay region in Fig. 2, and for the other regions in Supplement A.

Figure 3 illustrates the method for finding the dates of spring retreat and fall advance in Baffin Bay in one particular year (2005). The daily sea-ice area (gray curve) exhibits small daily fluctuations that can be attributed to the uncertainty in the underlying sea-ice concentration data. We smooth the daily values with a low-pass Gaussian-shaped filter in which 87% of the weight is within ±1 week of the central value (black curve). Then, starting from the minimum sea-ice area in summer, we search forward and backward in time for the first intersections of the smoothed time series with the threshold. The backward search gives the spring date (red vertical line) and the forward search gives the fall date (blue vertical line).

Occasionally the smoothed sea-ice area time series may cross the threshold more than once in spring and/or fall. Our method always chooses the crossing date that is closest in time to the summer minimum. In practice, out of 2736 crossing dates (36 years $\times$ 2 seasons $\times$ 19 regions $\times$ 2 time series per region), only 131 dates (4.8%) had any potential for ambiguity. In more than 95% of the cases there was clearly only a single crossing date.



### 3.3 Summer sea-ice concentration

For each region we calculated the mean sea-ice concentration for June 1 – October 31 for each
        year, 1979-2014.  While it has already been established that the sea-ice concentration in every
        region of the Arctic except the Bering Sea is declining in every month of the year (e.g., Perovich
        and Richter-Menge, 2009), the winter sea-ice cover will likely continue to provide suitable polar
        bear habitat for at least several more decades (Wiig et al., 2015), whereas the summer sea-ice
cover may not.  A summer sea-ice metric, therefore, measures the change in polar bear habitat
        during the season when that habitat is most vulnerable to change.

### 3.4 Number of ice-covered days per year

We calculated the number of days per year that the sea-ice area in each polar bear region
        exceeded the threshold defined in Section 3.2 (i.e., 50% of the way from mean September to
        mean March sea-ice area).  For example, in Fig. 3, the sea-ice area in Baffin Bay was greater
        than the 50% threshold for 220 days in the year 2005.  This sea-ice metric was used as a measure
        of polar bear habitat in the IUCN Red List assessment of polar bears (Wiig et al., 2015).

### 4 Results

### 4.1 Sea-ice metrics

In all 19 regions, the date of spring sea-ice retreat is trending earlier, the date of fall sea-ice
        advance is trending later, the length of the summer season is increasing, the summer sea-ice
        concentration is decreasing, and the number of ice-covered days per year is decreasing, for the
        period 1979-2014 (Table 3).  Most of the trends are statistically significant.  Trends in the date of
        spring sea-ice retreat are on the order of $-3$ to $-9$ days decade$^{-1}$, with the largest trend ($-16$ days
decade$^{-1}$) in the Barents Sea.  Trends in the date of fall sea-ice advance are on the order of $+3$ to
        $+9$ days decade$^{-1}$, with the largest trend ($+18$ days decade$^{-1}$) again in the Barents Sea.  This
        means that over the 3½ decades of this study, the time interval from the date of spring retreat to



the date of fall advance has lengthened by 3 to 9 weeks in most regions, and by 17 weeks in the Barents Sea. The summer (June-Oct) sea-ice concentration is declining at a rate of −1 to −9

percent decade$^{-1}$, depending on region. The number of ice-covered days is declining in all regions at the rate of −7 to −19 days decade$^{-1}$, with larger trends in the Barents Sea and central Arctic Basin. Results for the shallow (< 300 m) portions of each region are similar (Table 4). Note that some regions consist almost entirely of shallow depths (see Fig. 1 and Table 1).

Figure 4 illustrates results for one region: Baffin Bay. (See Supplement B for similar

plots for other regions). Sea-ice retreat in spring is changing by −7.3 days decade$^{-1}$ (red) and sea-ice advance in fall is changing by +5.4 days decade$^{-1}$ (blue), both statistically significant. The time interval between the spring and fall transition dates is changing by +12.7 days decade$^{-1}$ (Fig. 5; see Supplement C for similar plots for other regions). The summer sea-ice concentration is changing by −4.1 percent decade$^{-1}$ (Fig. 6; see Supplement D for similar plots for other

regions). The number of ice-covered days is changing by −12.7 days decade$^{-1}$, which is the negative of the fall-minus-spring trend: the loss of every ice-covered day occurs between the time of spring sea-ice retreat and fall sea-ice advance.

We also calculated the number of ice-covered days based on a 15% threshold of sea-ice area, as illustrated in Figures 7 and 8 (see Supplement E for similar plots for other regions). The

15% and 50% thresholds intersect the annual cycle of sea-ice area at different levels and therefore contain information about the shape of the annual cycle. In Baffin Bay, the rate of decline in the number of ice-covered days is about the same for both thresholds (Fig. 8). However, in the Chukchi Sea region (Supplement E) the rate of decline is faster for the 15% threshold, meaning that the rise and fall of the annual cycle of sea-ice area is steepening, leading

to faster transitions between summer and winter sea-ice coverage. In the Barents Sea (Supplement E) the opposite is happening. Further analysis of changes in the shape of the annual cycle of sea-ice area is possible but is beyond the scope of the present study.

## 4.2 Correlation of de-trended dates


Figure 4 shows that there is year-to-year variability about the trend lines in the dates of spring sea-ice retreat and fall sea-ice advance. Subtracting out the trend lines leaves residuals. We calculated the correlation of the spring residuals with the fall residuals (Table 3, last column).





The correlation is negative in most regions, often significantly so. This means that an early

spring sea-ice retreat (relative to the trend line) tends to be followed by a late fall sea-ice advance

(relative to the trend line), and vice versa. The de-trended spring and fall dates for Baffin Bay

are shown in Fig. 9.

      The negative correlations are likely the result of the ice-albedo feedback: when sea ice

retreats earlier than average in spring, the ocean has more time to absorb heat from the sun. The

extra heat is stored in the upper ocean through the summer, and must be released to the

atmosphere in the fall before sea ice can begin to form, thus delaying fall freeze-up. Conversely,

a late spring sea-ice retreat prevents the ocean from absorbing as much heat, allowing sea ice to

form earlier in the fall. The correlation is not perfect because other factors contribute to the dates

of sea-ice retreat and advance, such as short-term weather events and long-term climate patterns.

230          In regions with a strong negative correlation, this suggests a method for predicting the

date of fall sea-ice advance, once the date of spring sea-ice retreat has been observed: (1) Find

the slope ($S$) of the least-squares fit of the de-trended fall dates vs. the de-trended spring dates (as

in Fig. 7, red line). (2) Calculate the projected date of retreat ($D_r$) and date of advance ($D_a$) for

the current year by extrapolating the historical trends (Table 3). (3) In the current year, once the

date of spring sea-ice retreat has been observed ($D_r^{obs}$), predict the date of fall sea-ice advance as:

$D_a + S \times (D_r^{obs} - D_r)$. This is the date projected by the trend line plus the anomaly predicted by

the historical correlation of the spring and fall dates. This method should give several months of

lead time for the predicted date of fall sea-ice advance, with a higher degree of skill than simply

predicting a continuation of the fall linear trend, in those regions where the spring and fall dates

are significantly correlated.

## 4.3 Spatial patterns

The spatial pattern of trends in the date of spring sea-ice retreat (Fig. 10) shows that all trends

over shallow depths are statistically significant except in the eastern and southeastern Beaufort

Sea (in agreement with Steele et al., 2015) and in the north-central Canadian Arctic Archipelago.

Otherwise, the continental shelves around the Arctic show significantly earlier spring retreat,

generally $-3$ to $-9$ days decade$^{-1}$, with faster retreat in the northern Chukchi and East Siberian

seas, Kane Basin, and especially the Barents Sea. For the date of fall sea-ice advance (Fig. 11),





all regions have positive trends, but the trends are not statistically significant in some parts of the
Canadian Arctic Archipelago.  The rest of the continental shelf regions around the Arctic show
significantly later fall advance, generally 3 to 9 days decade$^{-1}$, with larger rates in the northern
Chukchi and East Siberian seas and in the Barents Sea, similar to the spring pattern.  The
increase in the length of the summer season (Fig. 12) shows the same pattern, with roughly

double the rate (since it equals the fall rate minus the spring rate).

Note that in this analysis, the Chukchi Sea region (CS) extends south of Bering Strait into
the northern Bering Sea.  We know from other analyses (e.g. Laidre et al., 2015a; Parkinson,
2014) that there has been a slight *increase* in sea ice in the Bering Sea.  Therefore the negative
trends for the Chukchi Sea reported here, while still statistically significant, are relatively small

because of the inclusion of the northern Bering Sea within the Chukchi Sea region.  Similarly,
the trends for the Arctic Basin region (AB) are relatively large because that region includes the
northern Chukchi Sea, where summer sea ice has been rapidly disappearing (e.g. Frey et al.,
2015; Parkinson, 2014).

**4.4 Sensitivity to threshold**

The calculation of the spring and fall transition dates is based on a sea-ice area threshold that is
halfway between the mean September sea-ice area and the mean March sea-ice area for each
region.  Different thresholds would lead to different transition dates.  How sensitive are the

transition dates to the actual choice of threshold?  The answer can be seen in Fig. 2 (and
Supplement A).  The rate of change of sea-ice area (i.e., its slope) is relatively steep at the times
of threshold crossing, indicating that sea ice diminishes quickly in spring, and grows back
quickly in fall, compared to the rate of change in winter and summer.  Therefore the transition
dates are relatively insensitive to the threshold, in the sense that a small change in the threshold

would lead to a small change in the transition dates.




## 5 Discussion

### 5.1 Previous studies of the timing of Arctic sea-ice advance and retreat

Our methods in this study are based on our previous work. Laidre et al. (2015a) calculated the
timing of sea-ice advance and retreat in 12 Arctic regions (1979-2013) for the Conservation of
Arctic Flora and Fauna (CAFF) Arctic Biodiversity Assessment (ABA). Laidre et al. (2015b)
focused on polar bear habitat in East Greenland, including changes in the timing of sea-ice
advance and retreat. Laidre et al. (2012) examined narwhal sea-ice entrapments and the timing
of fall sea-ice advance in six narwhal summering areas of Baffin Bay. Heide-Jørgensen et al.
(2012) considered changes in the timing of spring sea-ice retreat in the North Water Polynya.
All these studies found trends toward earlier spring sea-ice retreat and later fall sea-ice advance
from the 1980s to present.

    Our sea-ice metrics are currently being used in the IUCN PBSG Status Table
(http://pbsg.npolar.no/en/status/status-table.html), the primary source of scientific information
for managers, non-governmental organizations, and the public on the status of the world's polar
bears. The Status Table includes trends in the dates of spring sea-ice retreat, fall sea-ice
advance, and summer (June-Oct) sea-ice concentration for each of the 19 polar bear
subpopulations, as reported here, and will be updated accordingly. We also conducted a sea-ice
analysis for the IUCN Red List assessment of polar bears (Wiig et al., 2015) in which we
calculated the number of days per year (1979-2014) that the sea-ice area in each PBSG region
exceeded a threshold area (using the same threshold as in the present study).

    Other researchers have considered changes in the timing of sea-ice advance and retreat
without specific emphasis on polar bears. Stammerjohn et al. (2012) used daily sea-ice
concentration from satellite passive microwave data (1979-2007) to calculate trends in the dates
of sea-ice retreat and advance at every $25 \times 25$ km grid cell. Then they identified two regions
where the trends were particularly large, encompassing parts of the East Siberian / Chukchi /
Beaufort seas, and the Kara / Barents seas. Dates of sea-ice retreat in these regions trended
earlier by 15-18 days decade$^{-1}$, and dates of sea-ice advance trended later by 10-13 days
decade$^{-1}$, with correlations of de-trended dates on the order of $-0.8$. Their results are slightly



more extreme than ours (Table 3) because their regions were specifically tailored to include the largest trends, but our results are nevertheless generally consistent with theirs.

Parkinson (2014) used daily passive microwave data (1979-2013) to calculate and map the number of days per year with sea-ice concentration ≥ 15%, finding that most of the Arctic

seasonal ice zone (roughly all regions in Fig. 1 except AB) is experiencing a loss of 10-20 days decade$^{-1}$, with the most rapid loss in the Barents Sea. They also found that the trends are not sensitive to the 15% threshold, with similar trends obtained using 50%. The results are consistent with ours (Table 3) that show an increase in the number of ice-covered days.

Frey et al. (2015) used daily passive microwave data (1979-2012) to study the timing of

sea-ice break-up, freeze-up, and persistence in the Beaufort, Chukchi, and Bering seas, finding trends toward earlier break-up and later freeze-up in the Beaufort and Chukchi seas, with steeper trends since 2000. They also used wind and air temperature data to determine that for the localized areas that are experiencing the most rapid shifts in sea ice, those in the Beaufort Sea are primarily wind driven, while those offshore in the Canada Basin are primarily thermally driven.

Steele et al. (2015) looked at the timing of sea-ice retreat in the southeast (SE) and southwest (SW) Beaufort Sea using daily sea-ice concentration data (1979-2012). They found no trend in the date of retreat in the SE Beaufort Sea, but a trend toward earlier retreat in the SW Beaufort Sea. Furthermore, an increase in monthly mean easterly winds of ~1 m s$^{-1}$ during spring was associated with an earlier summer sea-ice retreat of 6-15 days, offering predictive

capability of sea-ice retreat with 2-4 months lead time.

Many studies in the last ten years have considered changes in the timing of sea-ice advance and retreat in the context of polar bear ecology. Stirling and Parkinson (2006) used daily sea-ice concentration from satellite passive microwave data to calculate the date of sea-ice break-up (50% concentration) in the spring in Baffin Bay for each year from 1979 through 2004,

finding a statistically significant trend toward earlier break-up ($-6.6 \pm 2.0$ days decade$^{-1}$). The timing of polar bear onshore arrival in western Hudson Bay was previously shown to be significantly related to the 50% sea-ice concentration threshold (Stirling et al., 1999). Other studies of sea-ice timing and polar bears include (see Table 2) Regehr et al. (2007), Obbard et al. (2007), Hamilton et al. (2014), Lunn et al. (2014), Laidre et al. (2015a), and Obbard et al. (2016).





### 5.2 Variability in the timing of sea-ice advance and retreat

Some regions such as East Greenland (EG) have high year-to-year variability (with respect to the

trend line) in the dates of sea-ice advance and retreat, while other regions such as Foxe Basin (FB) have low variability. The high variability is likely due to advection of sea ice through the region due to wind and currents, while the low variability indicates a lack of such advection, as noted by Laidre et al. (2012), who found that three sheltered sites on the western side of Baffin Bay had low variability in fall freeze-up dates, while sites near the North Water Polynya in

northern Baffin Bay, and in the East Greenland Current, had high variability. In regions where sea-ice advance and retreat are primarily driven by thermodynamics, the year-to-year variability will be lower than in regions where wind and currents are strong.

### 5.3 Correlation of dates of sea-ice retreat and advance


The negative correlations between the de-trended dates of sea-ice retreat and advance (Tables 3 and 4) are likely the result of the ice-albedo feedback, noted also by Stammerjohn et al. (2012). This is discussed in more detail by Blanchard et al. (2011), who attributed the "re-emergence of memory" in the fall to the several-month persistence of sea surface temperatures (SSTs),

enhanced by the ice-albedo feedback. We calculated the correlation of the date of fall sea-ice advance in year $n$ with the date of spring sea-ice retreat in year $n+1$, but the correlation was not significant in any region, suggesting that SST anomalies do not persist through the winter.

### 5.4 Sea-ice area vs. extent


Some sea-ice studies use sea-ice extent, rather than sea-ice area, to characterize sea-ice coverage. Sea-ice extent is the total area of all grid cells with sea-ice concentration greater than 15%, i.e., *not* weighted by the sea-ice concentration. If the sea-ice concentration in a grid cell exceeds 15%, the entire area of the grid cell counts toward the sea-ice extent. This is useful in some

contexts, but we believe that sea-ice area is a better measure of how much usable sea ice is actually present for polar bears. Also, sea-ice extent is a highly non-linear function of sea-ice concentration, which leads to more abrupt jumps in its time series than sea-ice area.



### 5.5 Melt onset and freeze-up


Some investigators have approached the idea of seasonal transitions in the Arctic by examining the dates of melt onset in the spring and freeze-up in the fall, based on the presence of liquid water in the surface layer of the ice (or snow) (Winebrenner et al., 1994 and 1996; Smith, 1998; Belchansky et al., 2004; Markus et al., 2009; Stroeve et al., 2014). In these studies, melt onset

and freeze-up are closely tied to the surface air temperature, but they are not indicators of sea-ice coverage or condition. For example, at the SHEBA station in the Beaufort Sea in 1997-1998 (Perovich et al., 1999), melt onset occurred on May 29 when rain fell, but the sea ice did not actually break up until the end of July when a storm passed through. Similarly in fall, melt ponds on the surface of the ice began to freeze in mid-August but the sea ice did not actually

consolidate into winter-like pack ice until early October (Stern and Moritz, 2002). Melt onset and freeze-up dates are useful as climate metrics, but for ice-dependent marine mammals, transition dates between seasons are best measured by the sea-ice coverage itself, rather than proxies tied to air temperature.

### 5.6 National Climate Assessment

The National Climate Assessment (NCA) summarizes the impacts of climate change across the United States, now and into the future, with the goal of better informing public and private decision-making at all levels. The third NCA report was released in May 2014 (Melillo et al.,

2014). It documents the decline of Arctic sea-ice extent, thickness, and volume, but not changes in the timing of sea-ice advance and retreat. One of the motivations of the present study was to develop a sea-ice climate metric (or indicator) with relevance to marine mammals that could be used in future NCA reports. The timing of sea-ice advance and retreat satisfies all the qualifications for climate indicators put forward by the NCA (NCA, 2011).


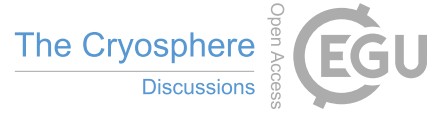

## 6 Conclusions


It is well established that the area of Arctic sea ice is declining in all months of the year, based on satellite passive microwave data from 1979 to the present (Sea Ice Index, 2016; IPCC, 2013). In this study we looked instead at the timing of sea-ice retreat in spring and advance in fall, because the duration of the sea-ice season (or equivalently the ice-free season) is important for

polar bears.

The PBSG recognizes 19 subpopulations of polar bears in 19 distinct regions of the Arctic. We have shown that over the course of the satellite record there has been a consistent and large loss of habitat for polar bears across the Arctic. In 17 of the 19 regions there are significant trends toward earlier spring sea-ice retreat. Most of the trends range from $-3$ to $-9$

days decade$^{-1}$, with the Arctic Basin ($-12$ days decade$^{-1}$) and the Barents Sea ($-16$ days decade$^{-1}$) the most extreme. In 16 of the regions there are significant trends toward later fall sea-ice advance. Most of the trends range from $+3$ to $+9$ days decade$^{-1}$, with the Arctic Basin ($+15$ days decade$^{-1}$) and the Barents Sea ($+18$ days decade$^{-1}$) again the most extreme. Over the 3½ decades of this study, the time interval from the date of spring retreat to the date of fall advance

has lengthened by 3 to 9 weeks in most regions, and by 17 weeks in the Barents Sea.

Global Climate Models (GCMs) predict ice-free Arctic summers by mid-century or sooner (IPCC, 2013; Overland and Wang, 2013). Spring sea-ice retreat will continue to arrive earlier and fall sea-ice advance will continue to arrive later, with no reversal in sight. Barnhart et al. (2015) used daily sea-ice output from a 30-member GCM ensemble, driven by the business-

as-usual emissions scenario (RCP 8.5), to map the annual duration of open water in the Arctic through 2100. They found that by 2050, the entire Arctic coastline and most of the Arctic Ocean will experience an additional one to two months of open water per year, relative to present conditions, which is consistent with extrapolation of the trends in Table 3.

What are the implications of these physical changes for the global population of polar

bears? Their dependence on sea-ice means that climate warming poses the single most important threat to their persistence (Obbard et al., 2010). Changes in sea ice have been shown to impact polar bear abundance, productivity, body condition, and distribution (Stirling et al., 1999; Durner et al., 2009; Regehr et al., 2010; Rode et al., 2012 and 2014; Bromaghin et al., 2015; Obbard et al., 2016). Furthermore, population and habitat models predict substantial



declines in the distribution and abundance of polar bears in the future (Durner et al., 2009;
Amstrup et al., 2008; Castro de la Guardia et al., 2013; Hamilton et al., 2014). This study offers
standardized metrics with which to compare polar bear habitat change across the 19
subpopulations, and provides a starting point for including sea-ice habitat change in circumpolar
polar bear management and conservation plans.


**Author contribution**

Stern carried out the sea-ice calculations in consultation with Laidre; Stern and Laidre prepared
the manuscript.


**Acknowledgements**

This work was supported by NASA under the programs *Development and Testing of Potential
Indicators for the National Climate Assessment*, grant NNX13AN28G (PI: H. Stern), and

*Climate and Biological Response*, grant NNX11A063G (PI: K. Laidre). We also acknowledge
support from the Greenland Institute of Natural Resources. We thank the National Snow and Ice
Data Center in Boulder for sea-ice concentration data, and NOAA for bathymetry data
(ETOPO1). We thank Eric Regehr, Steve Amstrup and Ceclia Bitz for conversations about sea-
ice metrics.




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





**Tables**

**Table 1**. Polar bear subpopulation region names, abbreviations, and areas.  See Fig. 1 for a map
of the regions.  The area of each region includes the marine portion only, not land.  The Number
of Cells is the number of SSM/I grid cells.  The Percent of Total Area is with respect to *All*
*regions* (last row).  The percent of area shallower than 300 m and deeper than 300 m are given in
the last two columns.  The *Pole hole* (second to last row) is the circular area around the North
Pole excluded from analysis due to the satellite orbits.  The Arctic Basin region (AB) surrounds
the pole hole but does not include it.  *All regions* includes all 19 subpopulation regions plus the
pole hole.


| Abbreviation | Subpopulation | Number of Cells | Area ($10^3$ km²) | % of Total Area | % < 300 m | % > 300 m |
|---|---|---|---|---|---|---|
| KB | Kane Basin | 81 | 53 | 0.3 | 68 | 32 |
| BB | Baffin Bay | 1042 | 656 | 4.3 | 28 | 72 |
| LS | Lancaster Sound | 380 | 243 | 1.6 | 73 | 27 |
| NB | Norwegian Bay | 108 | 70 | 0.5 | 84 | 16 |
| VM | Viscount Melville | 157 | 101 | 0.7 | 64 | 36 |
| NB | Northern Beaufort | 1055 | 677 | 4.4 | 23 | 77 |
| SB | Southern Beaufort | 529 | 333 | 2.2 | 59 | 41 |
| MC | M'Clintock Channel | 224 | 140 | 0.9 | 100 | 0 |
| GB | Gulf of Boothia | 100 | 62 | 0.4 | 99 | 1 |
| FB | Foxe Basin | 883 | 528 | 3.4 | 97 | 3 |
| WH | Western Hudson Bay | 326 | 188 | 1.2 | 100 | 0 |
| SH | Southern Hudson Bay | 744 | 417 | 2.7 | 100 | 0 |
| DS | Davis Strait | 2416 | 1367 | 8.9 | 40 | 60 |
| EG | East Greenland | 2237 | 1387 | 9.0 | 27 | 73 |
| BS | Barents Sea | 2379 | 1540 | 10.0 | 63 | 37 |
| KS | Kara Sea | 1645 | 1054 | 6.9 | 87 | 13 |
| LS | Laptev Sea | 2169 | 1393 | 9.1 | 84 | 16 |
| CS | Chukchi Sea | 1840 | 1117 | 7.3 | 98 | 2 |
| AB | Arctic Basin | 4307 | 2813 | 18.3 | 15 | 85 |
|  | Pole hole | 1799 | 1193 | 7.8 | 0 | 100 |
|  | All regions | 24421 | 15332 | 100.0 | 50 | 50 |



**Table 2**. Recent literature where sea-ice metrics were used for analysis of polar bear habitat. Note that these studies examined habitat for a single polar bear subpopulation (or geographically close set of subpopulations). Abbreviations: PM (Passive Microwave), SIC (sea-ice concentration), CIS (Canadian Ice Service).

| Subpopulation | Data | Years | Methods for sea-ice metric | Reference |
|---|---|---|---|---|
| Western Hudson Bay | Daily PM SIC | 1979-2004 | Calculated daily percent sea-ice cover in the region. **Date of spring sea-ice break-up** = date when ice cover fell below 50% | Stirling & Parkinson (2006) |
| Western Hudson Bay | Daily PM SIC | 1984-2004 | **Date of spring sea-ice breakup** = date when ice cover fell below 50% (same as Stirling and Parkinson, 2006). | Regehr et al. (2007) |
| Southern Hudson Bay | Daily PM SIC | 1984-2003 | **Date of spring sea-ice break-up** = date when ice cover fell below 50% (same as Stirling and Parkinson, 2006). **Date of fall sea-ice freeze-up** = date when ice cover rose above 50% **Ice-free period** = number of days between break-up and freeze-up | Obbard et al. (2007) |
| Southern Beaufort Sea | Daily PM SIC | 2001-2005 | Calculated the daily percent sea-ice cover for the continental shelf only (depth < 300 meters). **Number of ice-free days** = number of days per calendar year with ice cover < 50% | Regehr et al. (2010) |
| Northern Beaufort Sea | Daily PM SIC | 1979-2006 | **Mean annual number of grid cells with sea-ice concentration > 50%**; calculated for continental shelf only (depth < 300 meters); excluding a buffer of one ocean grid cell along all coastlines. Second sea ice covariate derived from the **resource selection functions** (RSFs) of Durner et al. (2009). | Stirling et al. (2011) |
| Baffin Bay, Davis Strait | Mean weekly SIC (CIS) | 1977-2010 | **Mean weekly sea-ice concentration** from 15 May to 15 October. | Rode et al. (2012) |
| Chukchi Sea, Southern Beaufort Sea | Daily PM SIC | 1985-1993, 2007-2010 | **Reduced-ice days per year** = number of days with sea-ice area < 6250 km$^2$ (continental shelf of each region only, depth < 300 meters) **Distance to ice edge** = daily minimum distance from continental shelf to pack ice, averaged over all days in September. When pack ice is over the continental shelf the distance is set to zero. | Rode et al. (2014) |
| Baffin Bay | Daily PM SIC | 1979-2009 | **Sea-ice concentration in April, May, June** for the continental shelf only (depth < 300 meters). (Note that the continental shelf consists of 2 parts: Baffin Island in the west and Greenland in the east). | Peacock et al. (2012) |
| Davis Strait | Mean weekly SIC (CIS) | 1974-2007 | **Mean weekly sea-ice concentration** from 14 May to 15 October | Peacock et al. (2013) |
| Canadian Arctic Archipelago | MIT General Circulation Model (GCM) | 2006-2100 | Future projections of sea ice were made using the MIT GCM with 18-km grid size and monthly output, forced by "business as usual" RCP8.5 emission scenario. **Month of spring sea-ice breakup** = the first month in a given year with sea-ice concentration < 50% **Month of fall sea-ice freeze-up** = the first month after | Hamilton et al. (2014) |



| | | | break-up with sea-ice concentration ≥ 10% **Ice-free season** = the time from break-up to freeze-up. If all months of the year have sea-ice concentration < 10% then the ice-free season is 12 months. | |
|---|---|---|---|---|
| Western Hudson Bay | Daily PM SIC | 1979-2012 | Calculated daily percent sea-ice cover in the region. **Date of spring sea-ice break-up** = date when ice cover fell below 50% (same as Stirling and Parkinson, 2006) and stayed below 50% for at least 3 consecutive days **Date of fall sea-ice freeze-up** = date when ice cover rose above 50% and stayed above 50% for at least 3 consecutive days **Ice decay** = rate of sea-ice loss from 1 May until the date of complete disappearance of sea ice; calculated as the absolute value of the slope of the ordinary least squares regression line of ice concentration vs. time | Lunn et al. (2014) |
| East Greenland | Daily PM SIC | 1979-2012 | Calculated the daily sea-ice area in the region. Defined threshold area A = halfway between mean March ice area and mean September ice area, where the means are calculated over the baseline period 1979-1988. **Date of spring sea-ice break-up** = date when ice area fell below threshold area A **Date of fall sea-ice freeze-up** = date when ice area rose above threshold area A | Laidre et al. (2015a) |
| Chukchi Sea, Southern Beaufort Sea | Daily PM SIC | 1979-2013 | Calculated the daily sea-ice area in each region. Defined threshold area A = halfway between mean March ice area and zero area, where the mean March area is calculated over the baseline period 1979-2013. **Ice-covered days** = number of days each year with ice area > threshold area A. Calculated the mean number of ice-covered days for 1994-2013 and then projected the number of ice-covered days forward in time. | Regehr et al. (2015) |
| Southern Beaufort Sea | | 2001-2010 | **Summer habitat** = sum of monthly indices of area of optimal polar bear habitat over continental shelf for July through Oct each year (from Durner et al., 2009). **Melt season** = time between melt onset and freeze onset ("inner melt length" from Stroeve et al., 2014). | Bromaghin et al. (2015) |
| Southern Hudson Bay | Daily PM SIC | 1980-2012 | **Date of spring sea-ice break-up** = date when mean ice concentration falls below 5% **Date of fall sea-ice freeze-up** = date when mean ice concentration rises above 5% | Obbard et al. (2016) |






**Table 3**. Trend in date of spring sea-ice retreat (days decade$^{-1}$); trend in date of fall sea-ice advance (days decade$^{-1}$); trend in length of summer season (days decade$^{-1}$); trend in June-October sea-ice concentration (percent concentration decade$^{-1}$); trend in number of ice-covered days (days decade$^{-1}$); and correlation of de-trended dates of spring retreat and fall advance (dimensionless). All quantities are computed from the total marine area of each region, regardless of depth (compare Table 4), for the period 1979-2014. The trend in the length of the summer season (Fall–Sp Trend) is equal to the fall trend minus the spring trend. Statistical significance is indicated by * (95% level) or ** (99% level) according to a two-sided F test (for trends) or a two-sided t test (for correlations).

| Subpopulation | Spring Trend | | Fall Trend | | Fall–Sp Trend | | Jun-Oct Ice Con | | Ice-Cov Days | | Corr of Dates | |
|---|---|---|---|---|---|---|---|---|---|---|---|---|
| Kane Basin | -6.8 | * | 5.6 | ** | 12.4 | ** | -5.4 | ** | -14.1 | ** | -0.55 | ** |
| Baffin Bay | -7.3 | ** | 5.4 | ** | 12.7 | ** | -4.1 | ** | -12.7 | ** | -0.64 | ** |
| Lancaster Sound | -5.4 | * | 4.7 | ** | 10.1 | ** | -4.4 | ** | -10.6 | ** | -0.11 | |
| Norwegian Bay | -1.2 | | 4.3 | | 5.5 | | -1.6 | | -7.1 | * | -0.21 | |
| Viscount Melville | -4.0 | | 7.9 | | 11.8 | * | -4.7 | ** | -12.3 | ** | 0.36 | * |
| Northern Beaufort | -6.0 | * | 3.1 | | 9.0 | * | -4.3 | ** | -9.3 | * | -0.40 | * |
| Southern Beaufort | -9.0 | ** | 8.8 | ** | 17.8 | ** | -9.3 | ** | -17.5 | ** | -0.50 | ** |
| M'Clintock Channel | -4.1 | ** | 5.9 | ** | 10.0 | ** | -5.1 | ** | -11.1 | ** | -0.74 | ** |
| Gulf of Boothia | -8.6 | ** | 7.6 | ** | 16.2 | ** | -8.9 | ** | -18.6 | ** | -0.57 | ** |
| Foxe Basin | -5.3 | ** | 5.7 | ** | 11.0 | ** | -3.3 | ** | -11.4 | ** | -0.58 | ** |
| Western Hudson Bay | -5.1 | ** | 3.5 | ** | 8.7 | ** | -2.9 | ** | -8.6 | ** | -0.25 | |
| Southern Hudson Bay | -3.0 | * | 3.6 | * | 6.6 | ** | -1.8 | * | -6.8 | ** | -0.35 | * |
| Davis Strait | -7.4 | ** | 9.2 | ** | 16.6 | ** | -1.0 | ** | -17.1 | ** | -0.35 | * |
| East Greenland | -5.7 | ** | 4.8 | * | 10.5 | ** | -1.4 | * | -10.4 | ** | -0.34 | * |
| Barents Sea | -16.4 | ** | 18.2 | ** | 34.6 | ** | -3.8 | ** | -41.0 | ** | -0.46 | ** |
| Kara Sea | -9.2 | ** | 7.3 | ** | 16.5 | ** | -7.9 | ** | -16.9 | ** | -0.49 | ** |
| Laptev Sea | -6.9 | ** | 7.0 | ** | 13.9 | ** | -9.4 | ** | -13.5 | ** | -0.78 | ** |
| Chukchi Sea | -4.0 | ** | 5.3 | ** | 9.3 | ** | -4.0 | ** | -8.9 | ** | -0.39 | * |
| Arctic Basin | -11.9 | ** | 15.2 | ** | 27.1 | ** | -6.0 | ** | -24.6 | ** | 0.35 | * |





**Table 4**. Same as Table 3 but for the shallow (< 300 m) portions of each region.

| Subpopulation | Spring Trend | | Fall Trend | | Fall–Sp Trend | | Jun-Oct Ice Con | | Ice-Cov Days | | Corr of Dates | |
|---|---|---|---|---|---|---|---|---|---|---|---|---|
| Kane Basin | -9.7 | ** | 5.5 | ** | 15.2 | ** | -6.9 | ** | -15.1 | ** | -0.36 | * |
| Baffin Bay | -8.4 | ** | 9.7 | ** | 18.1 | ** | -3.3 | ** | -19.8 | ** | -0.54 | ** |
| Lancaster Sound | -7.6 | ** | 4.6 | ** | 12.2 | ** | -4.3 | ** | -11.2 | ** | -0.35 | * |
| Norwegian Bay | -1.3 | | 4.2 | | 5.5 | | -1.6 | | -7.0 | ** | -0.21 | |
| Viscount Melville | -4.3 | | 6.9 | | 11.2 | * | -4.3 | ** | -11.7 | ** | 0.31 | |
| Northern Beaufort | -5.6 | | 3.5 | ** | 9.1 | * | -3.6 | * | -8.5 | * | -0.62 | ** |
| Southern Beaufort | -7.3 | ** | 8.6 | ** | 15.9 | ** | -7.9 | ** | -15.5 | ** | -0.53 | ** |
| M'Clintock Channel | -4.1 | ** | 5.8 | ** | 10.0 | ** | -5.2 | ** | -11.0 | ** | -0.74 | ** |
| Gulf of Boothia | -8.6 | ** | 7.6 | ** | 16.2 | ** | -9.0 | ** | -18.8 | ** | -0.57 | ** |
| Foxe Basin | -5.2 | ** | 5.6 | ** | 10.9 | ** | -3.2 | ** | -11.3 | ** | -0.57 | ** |
| Western Hudson Bay | -5.1 | ** | 3.5 | ** | 8.7 | ** | -2.9 | ** | -8.6 | ** | -0.25 | |
| Southern Hudson Bay | -3.0 | * | 3.6 | * | 6.6 | ** | -1.8 | * | -6.8 | ** | -0.35 | * |
| Davis Strait | -6.9 | ** | 8.0 | ** | 14.9 | ** | -1.9 | ** | -14.7 | ** | -0.26 | |
| East Greenland | -4.5 | ** | 4.6 | ** | 9.0 | ** | -3.0 | * | -9.4 | ** | -0.30 | |
| Barents Sea | -17.0 | ** | 21.0 | ** | 37.9 | ** | -4.2 | ** | -44.6 | ** | -0.46 | ** |
| Kara Sea | -8.8 | ** | 7.0 | ** | 15.8 | ** | -7.3 | ** | -16.2 | ** | -0.47 | ** |
| Laptev Sea | -6.8 | ** | 6.5 | ** | 13.3 | ** | -9.1 | ** | -13.2 | ** | -0.77 | ** |
| Chukchi Sea | -4.1 | ** | 5.4 | ** | 9.5 | ** | -4.1 | ** | -9.1 | ** | -0.39 | * |
| Arctic Basin | -9.4 | ** | 16.8 | ** | 26.1 | ** | -9.0 | ** | -29.3 | ** | -0.18 | |



**Figures**


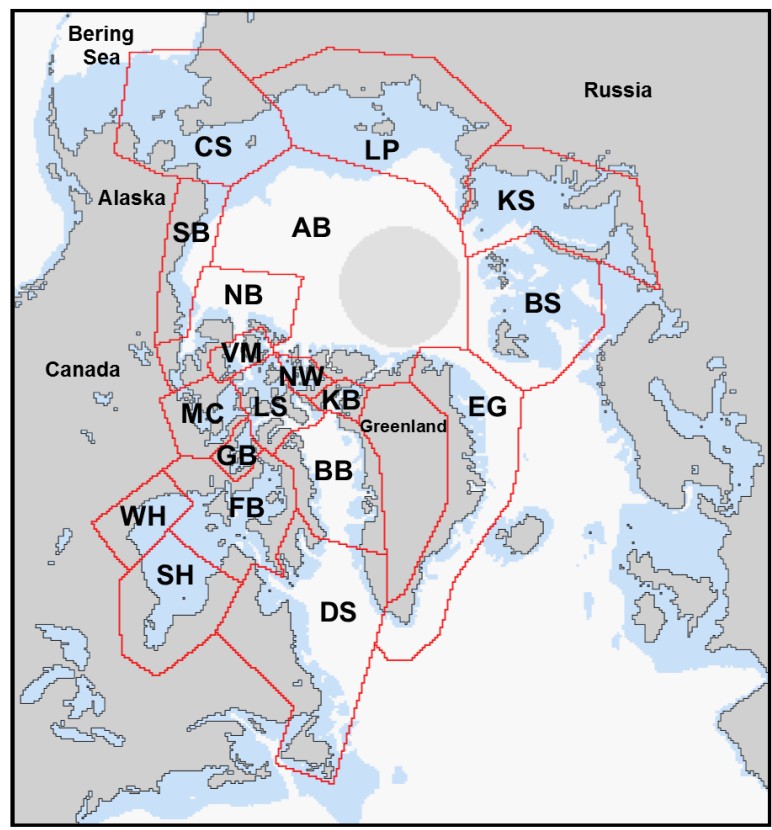

**Figure 1**. Map of the 19 PBSG polar bear subpopulation regions, with shallow depths (< 300 m) in blue. See Table 1 for subpopulation names corresponding to the abbreviations on the map.




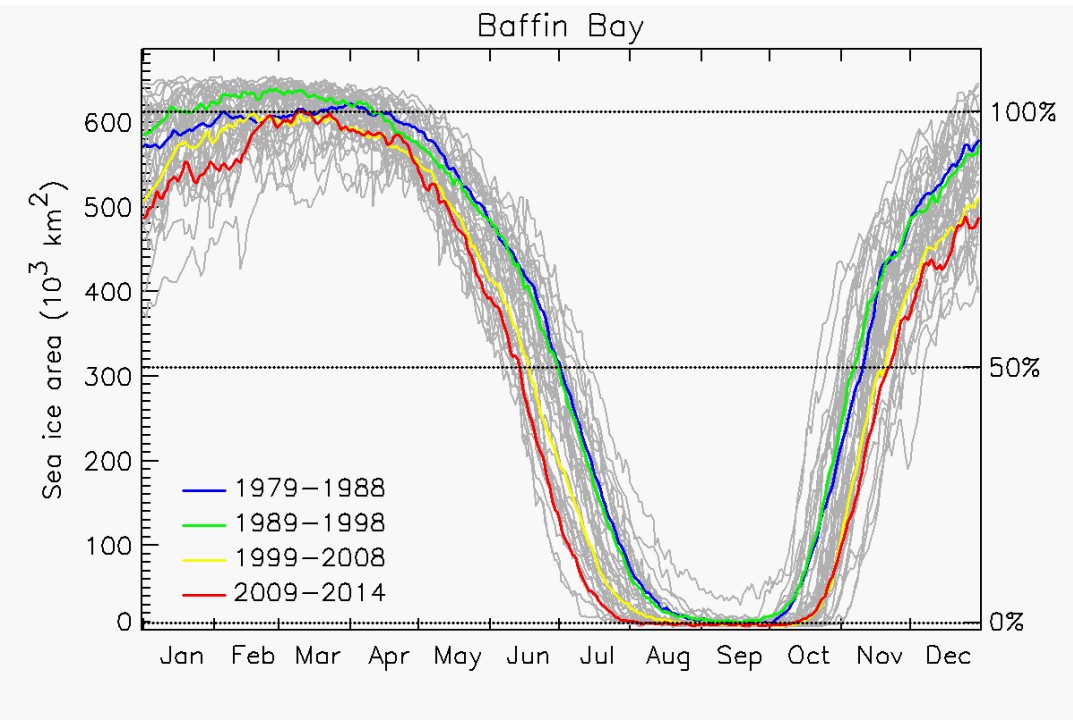


**Figure 2**. Daily sea-ice area in Baffin Bay (all depths), January-December, 1979-2014 (gray curves). The colored curves are decadal averages, as indicated in the legend. The upper horizontal dotted line (at $613 \times 10^3$ km$^2$) is the average sea-ice area in March (1979-2014); the lower horizontal dotted line (at $9 \times 10^3$ km$^2$) is the average sea-ice area in September. The
middle horizontal dotted line, halfway between the upper and lower lines, is the threshold for determining the spring and fall transition dates in Baffin Bay. See Supplement A for similar plots for other regions.






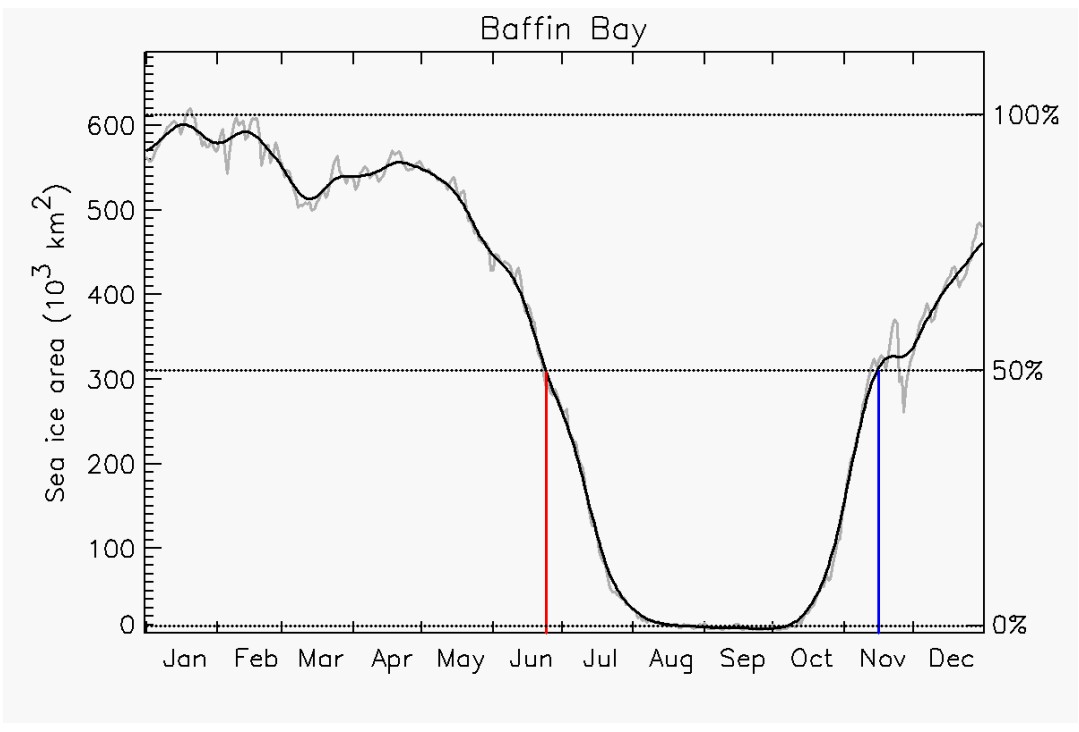

**Figure 3**. Determination of the spring and fall transition dates for the year 2005 in Baffin Bay. The gray curve is the daily sea-ice area; the black curve is a smoothed version. The horizontal dotted line (at $311 \times 10^3$ km$^2$) is the threshold. The intersection of the threshold with the
smoothed (black) curve determines the spring (red) and fall (blue) transition dates.



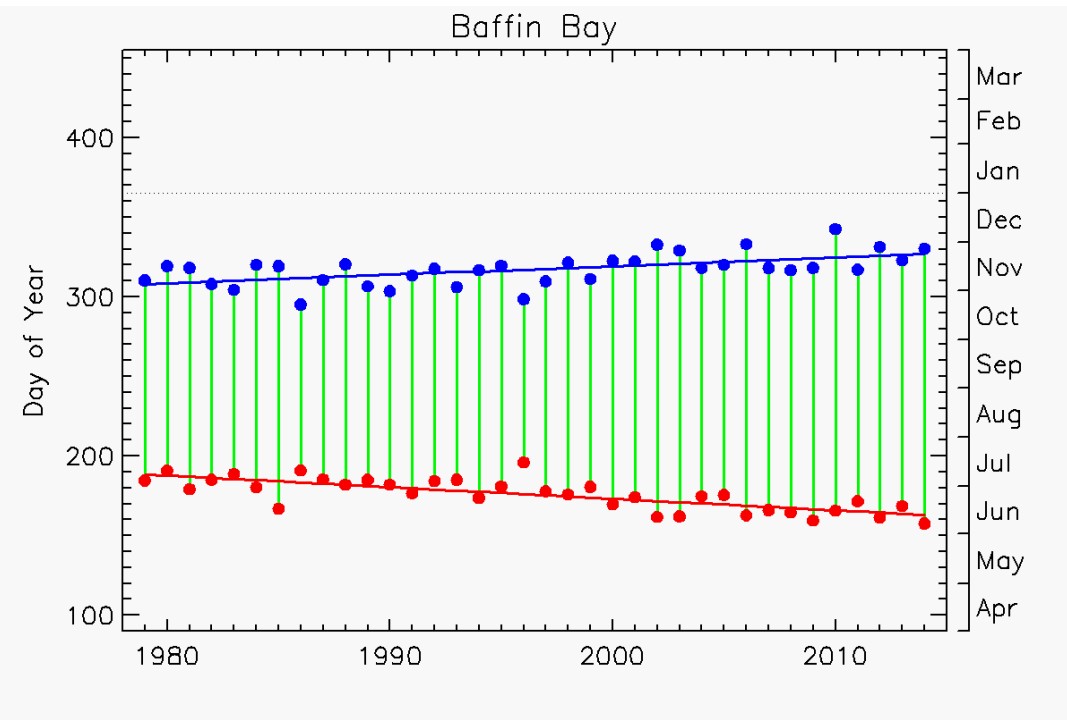

**Figure 4**.  Dates of sea-ice retreat (red) and sea-ice advance (blue) in Baffin Bay (all depths) for
         1979-2014.  The red and blue lines are least-squares fits.  The vertical green lines indicate the
         time interval between retreat and advance (i.e., length of summer season).  See Table 3 for
         trends.  See Supplement B for similar plots for other regions.






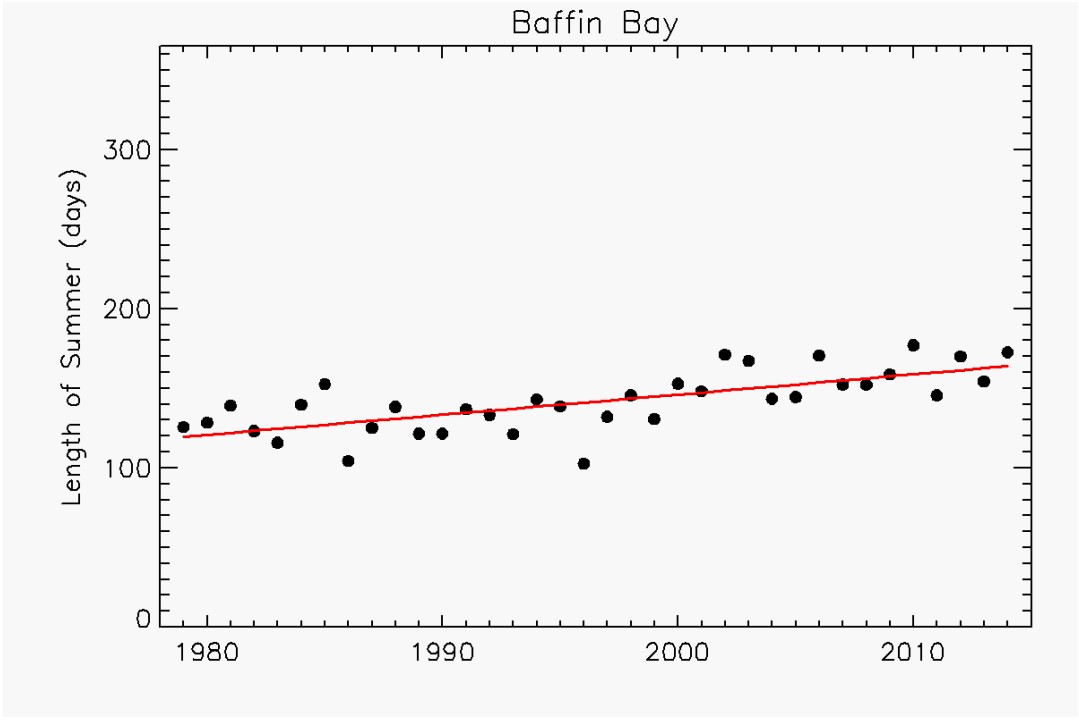

**Figure 5**. Length of the summer season (from spring sea-ice retreat to fall sea-ice advance) vs.
year for Baffin Bay (all depths), with least-squares line in red (slope: +12.7 days decade$^{-1}$). See
Supplement C for similar plots for other regions.



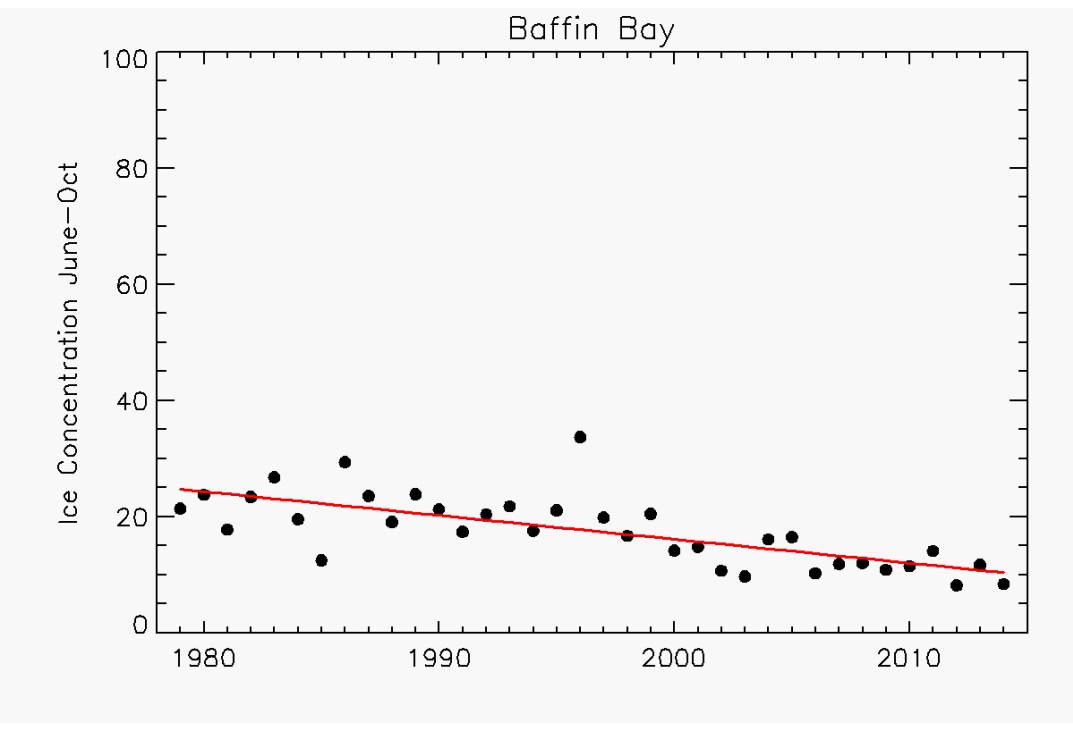


**Figure 6**. Summer (June through October) sea-ice concentration vs. year for Baffin Bay (all depths), with least-squares line in red (slope: −4.1 percent decade$^{-1}$). See Supplement D for similar plots for other regions.





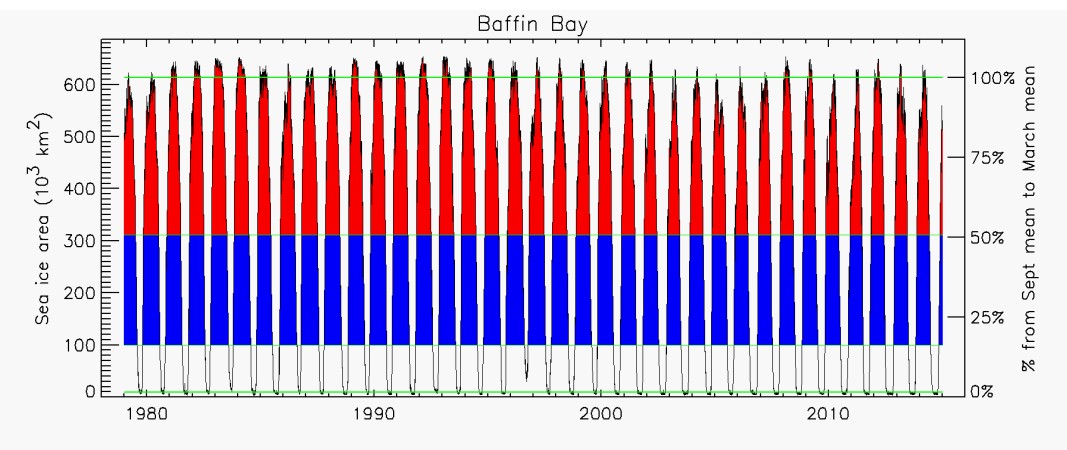

**Figure 7**. Sea-ice area in Baffin Bay (all depths), 1979-2014. Top green line is mean March sea-ice area; bottom green line is mean September sea-ice area. Two thresholds are shown: 15% and 50% of the way from the mean September area to the mean March area.

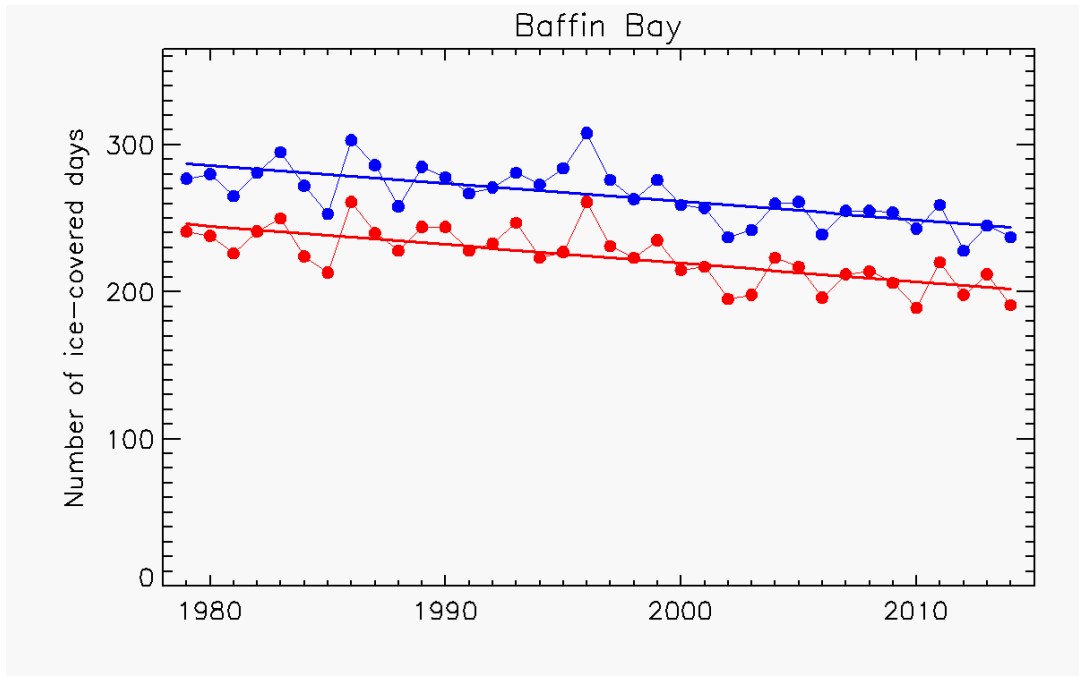


**Figure 8**. Number of ice-covered days in Baffin Bay (all depths), 1979-2014, based on two thresholds: 15% (blue) and 50% (red) (see also Fig. 7). Least-squares lines are also shown. See Supplement E for similar plots for other regions.

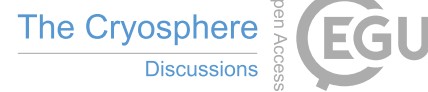


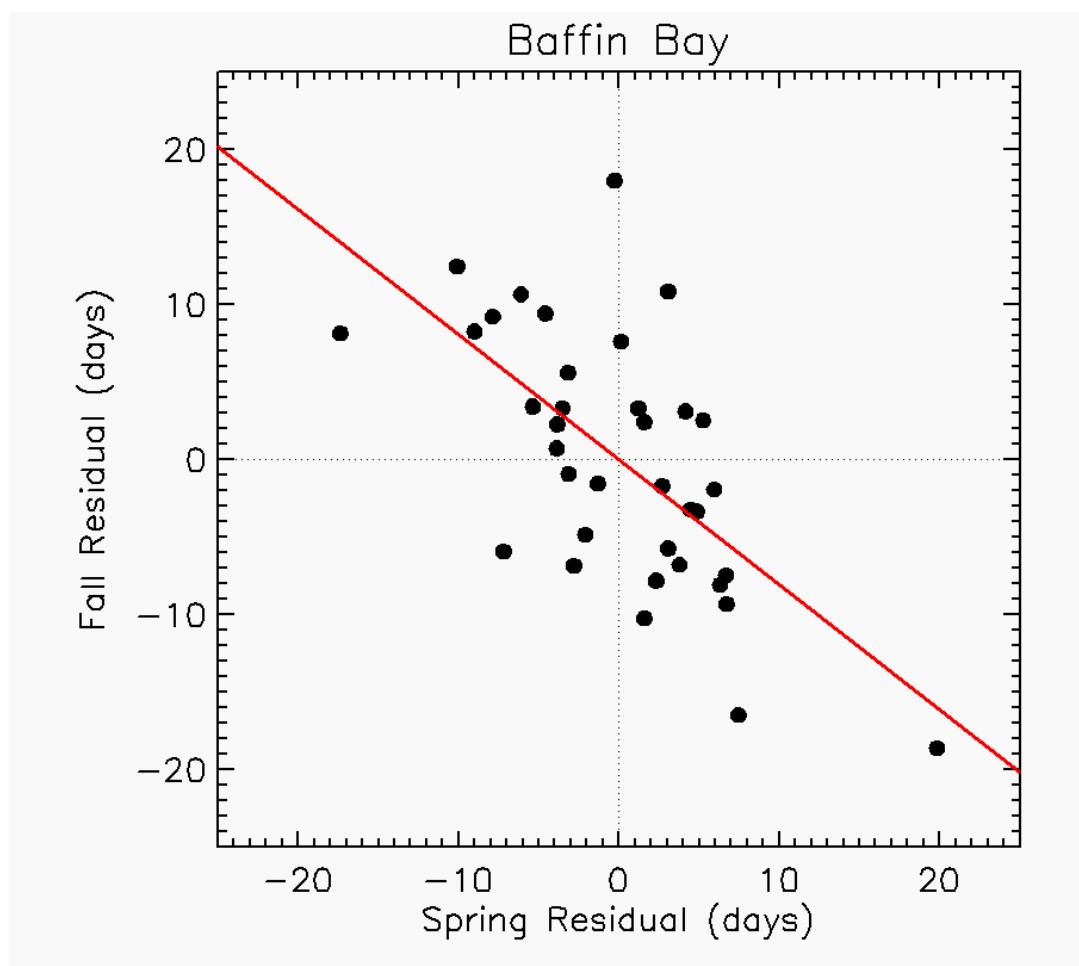

**Figure 9**. Date of fall sea-ice advance (de-trended) vs. date of spring sea-ice retreat (de-trended)
for Baffin Bay (all depths). The de-trended dates have correlation −0.64. This suggests that the
date of fall sea-ice advance can be predicted from the date of spring sea-ice retreat with more
skill than simply extrapolating the fall trend. See Table 3 for correlations in all regions. The red
line is the least-squares fit.





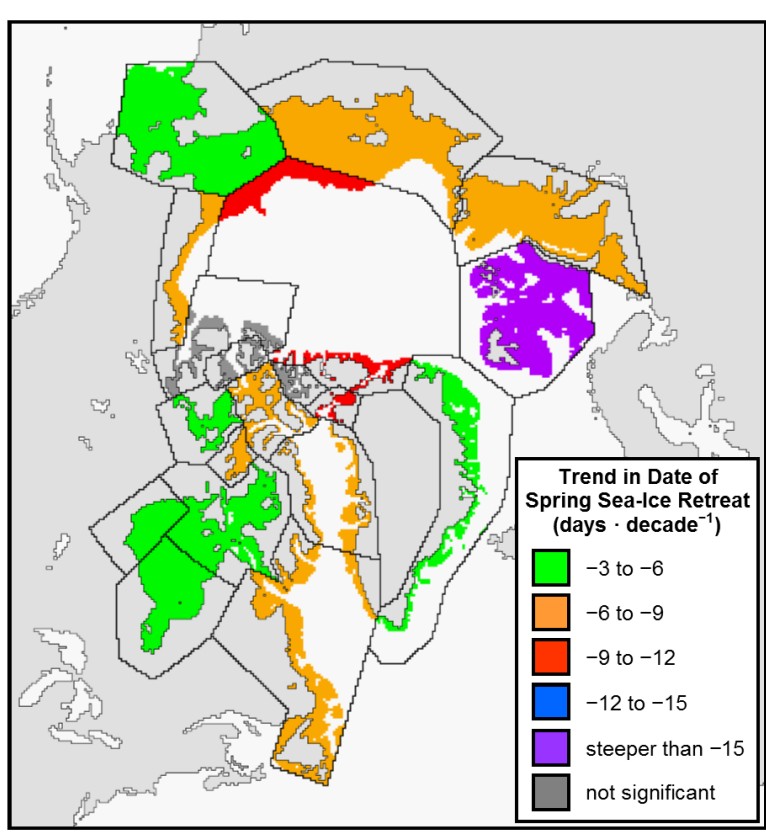


**Figure 10**. Trend map of the date of spring sea-ice retreat for the shallow parts of each PBSG region. Trends are also given in Table 4.





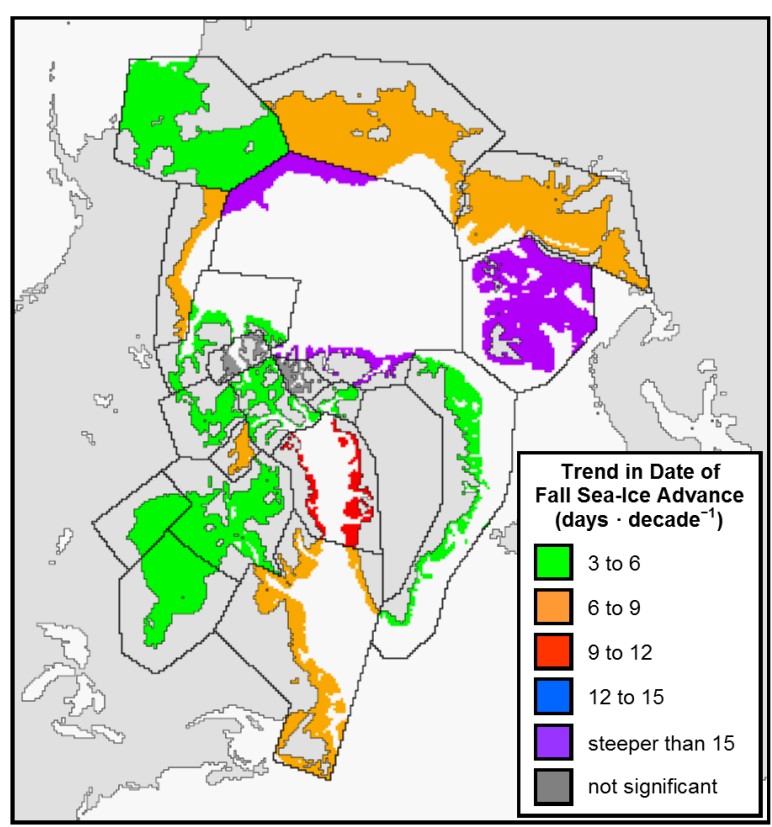


**Figure 11**. Trend map of the date of fall sea-ice advance for the shallow parts of each PBSG region. Trends are also given in Table 4.



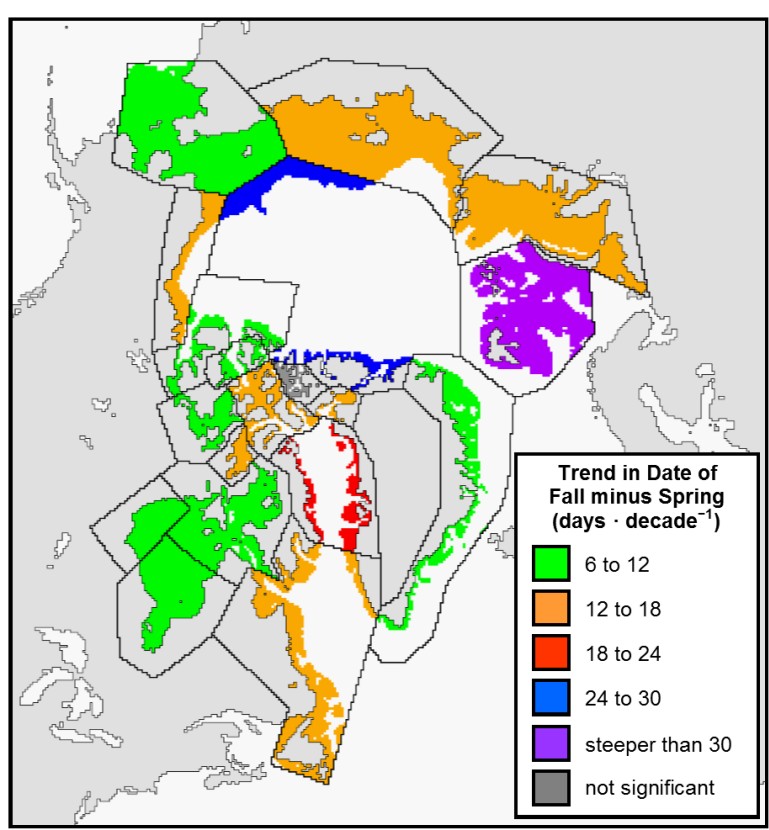


**Figure 12**. Trend map of the length of the summer season for the shallow parts of each PBSG region. Trends are also given in Table 4.