# Peer review of "Sea-Ice Indicators of Polar Bear Habitat"

_The Cryosphere, 2016_

## Referee Comment (RC1) · Anonymous Referee #1 · 2 Jun 2016

I find the work to be both interesting and useful. The methods applied are robust and for the most part, clearly presented.

The introduction would benefit from some additional context and referencing. Use of the literature was acceptable but weak for many issues. Many will not be familiar with the issues being raised nor with polar bear ecology and thus, a fuller use of the literature and key points would be useful (e.g., what do polar bears hunt, do they migrate, mating season etc.). There is merit to a fuller examination of existing analyses and acknowledgment of scientific priority of the approach undertaken. Excess use of self-citation does not put the work, which is solid and useful, in context. Further, how this approach may (or may not) be useful to other taxa would be helpful. Further context for what the ice metrics may mean for polar bears would be useful. What are the possible consequences on reproduction, survival, or population trend. A couple of sentences would aid most readers understand the significance of the study, which is very polar

bear centric.

The manuscript would benefit from a clearer statement of the objectives.

13 – I would remove the term "distinct" – this cannot be meaningfully interpreted – further, the term distinct is not used on line 38-39

15 – it is somewhat simplistic to say that polar bear phenology is tied to sea ice. Their phenology is also linked to day length (circannual rhythms), physiological processes, etc. Tied sounds like cause-effect to me but perhaps just soften the wording and replace "tied".

58 – some further clarification would be useful here on the "need to develop" – the logic isn't particularly well developed and it may be worth referencing Vongraven, D. et al. 2012. A circumpolar monitoring framework for polar bears. Ursus Monograph 5:1-66.

Further, it would be useful to put this study in the context of other marine species. Do the proposed metrics work for seals, seabirds, whales etc.? As written, the work is narrowly focused on polar bears yet the applicability is broader and there would be great uptake and use of the work if it could be generalized a bit.

67 – use the primary literature Wiig et al. 2015 is a secondary citation.

72 – "behaviorally tied" – given that the journal's audience may not be familiar with the species, some details would be useful.

75-6 – provide citations – this is a well documented area and not an original or new idea

89-92 – SSM/I is well documented to perform poorly during break-up and freeze-up – this issue should be discussed more fully: in particular, bias in the data. I agree, however, that the bias is likely consistent across space and time so the trends are likely valid and thus, this is not a central concern for the manuscript's validity.

100-106 – the reasons for the bathymetry data are not clearly articulated. Why is

bathymetry data useful? It is only clear further down (line 115) but for flow, please introduce this in the objectives.

102 – sea ice is defined using 15% ice cover but many studies show that polar bears don't use ice cover < 30% or even 50%. Some justification from the literature is warranted. Further, ice cover during freeze-up is different that ice use during break up (Cherry, S.G. et al. 2013. Migration phenology and seasonal fidelity of an Arctic marine predator in relation to sea ice dynamics. Journal of Animal Ecology 82:912-921.)

121 – what the outlier discovery was interesting, it's not central to the study and the last sentences of this paragraph could be removed.

141 – The manuscript presents data for Baffin Bay yet no reason for this was given. The other populations are available in the appendix but the focus on Baffin Bay might warrant a bit more information about this population for background. This is not essential but for those unfamiliar with the polar bears there, it may be useful context (e.g., is this an area that is ice-free is summer or not).

183 – "Most of the trend are statistically significant." – could you give a % (x /76) (i.e., 4 metrics for 19 populations =76) OR for each of the metrics individually (preferred).

183-87 – some clarification of + and – mean might be useful. The – for retreat means earlier and advance means later but "advance" being a positive number made be stop and puzzle the result. You could simply put "(i.e., earlier)" and "(i.e., later)" to aid clarity.

223-229 – this is not a result but discussion. There are many sources that can be cited for this paragraph. This is not a new finding and presenting it as such ignores the literature.

285 – the present study is very similar in form and analysis to Parkinson, C.L. 2014. Spatially mapped reductions in the length of the Arctic sea ice season. Geophys Res Lett 41:4316-4322. doi:10.1002/2014GL060434. I suggest a more thorough comparison of the present manuscript and the published work is undertaken at the start of the

discussion. This sort of analysis has been conducted for many polar bear population and greatly predates The Laidre et al. and Heide-Jorgensen et al. works given priority in the discussion. A fuller coverage of the issue would be useful (i.e., move the cursory treatment of similar studies forward in the discussion for proper context). This field of study has been considered for many Arctic species well before 2012 and it is appropriate to acknowledge scientific priority at the start of the discussion (i.e., not half way through in passing).

289 – narwhal is missing its binomial name.

344-46 – What is mean by "low variability" or "high year-to-year variability" – was this measured (could CV be used or other method (SE, SD)).

397 – "relevance to marine mammals" – non-mammalogists may not have much insight to what species you are referring to. A brief list or examples might be useful (perhaps highlighting the most sensitive species and referencing other studies for this sensitivity).

406-420 – I found this repetitive and could be removed without loss.

430 Obbard et al. 2010 is gray literature – please use core peer-reviewed works to support this statement (of which there are many).
* * *

---

## Referee Comment (RC2) · Anonymous Referee #2 · 23 Jun 2016

This paper was very well written and easy to follow. It is solid work with sound analyses based on the best available data, and the results are striking and present material of high value that will be important as reference material and also as a basis for future work. I have no major questions or ideas for improvement of this article. Below a couple of minor comments.

Introduction, line 46: "The global population size is estimated to be about 25,000..)." I am not sure if "estimated" is the best wording here, as 25,000 is the sum of several estimates and guesses/extrapolations from areas with little or no data, so maybe rewrite.

Line 47: "Genetic analysis indicates that there is considerable gene flow between some subpopulations, while others are relatively discrete (….)." I would say "shows" rather than "indicates". Genetic structure is very low. Would also rewrite last part of sentence, maybe "…. It is more modest for some". I do not think subpopulations anywhere is

close to being discrete.

Methods, page 6, line 164: ". . .. The winter sea-ice cover will likely continue to provide suitable polar bear habitat for at least several more decades (. . .)." Where? In most places?

Discussion: page 11, line 318, ". . . that show an increase in the number of ice-covered days" Should be decrease, not increase? Or ice-free days, not ice-covered days?

---

## Author Comment (AC1) · 17 Aug 2016

I find the work to be both interesting and useful. The methods applied are robust and for the most part, clearly presented. The introduction would benefit from some additional context and referencing. Use of the literature was acceptable but weak for many issues. Many will not be familiar with the issues being raised nor with polar bear ecology and thus, a fuller use of the literature and key points would be useful (e.g., what do polar bears hunt, do they migrate, mating season etc.). There is merit to a fuller examination of existing analyses and acknowledgment of scientific priority of the approach undertaken. Excess use of self-citation does not put the work, which is solid and useful, in context. Further, how this approach may (or may not) be useful to other taxa would be helpful. Further context for what the ice metrics may mean for polar bears would be useful. What are the possible consequences on reproduction, survival, or population trend. A couple of sentences would aid most readers understand the significance of the study, which is very polar bear centric.

We thank the reviewer for all these ideas to improve the paper.

Regarding the Introduction, we have changed
"Polar bears depend on sea ice as a platform for hunting." to
"Polar bears depend on sea ice as a platform for hunting ice seals, their primary prey."
The next sentence, "Sea ice also facilitates their seasonal movements, mating, and, in some areas, maternal denning (Wiig et al., 2015)", touches on the importance of sea ice to polar bears, and refers the interested reader to Wiig et al. (2015) for a good introduction to polar bear ecology – the citation in the References section includes a URL (web link) to the report. While admittedly brief, the first paragraph is just meant to introduce the reader to the most basic information about polar bears, with references to more complete works.

Regarding "a fuller examination of existing analyses..." please see our response below to the comment of line 285.

Regarding "how this approach may (or may not) be useful to other taxa..." please see our response below to the second part of the comment of line 58.

Regarding "what the ice metrics may mean for polar bears..." please see the last paragraph of the paper, which begins with the sentence "What are the implications of these physical changes for the global population of polar bears?" We believe this paragraph already addresses the reviewer's comment.

The manuscript would benefit from a clearer statement of the objectives.

We have added the following sentence at the end of the second paragraph of the Introduction: "Thus the objective of this study is to propose and produce metrics of polar bear sea-ice habitat, which are also relevant to other Arctic marine mammal species."

13 – I would remove the term "distinct" – this cannot be meaningfully interpreted – further, the term distinct is not used on line 38-39

Deleted "distinct"

15 – it is somewhat simplistic to say that polar bear phenology is tied to sea ice. Their phenology is also linked to day length (circannual rhythms), physiological processes, etc. Tied sounds like cause-effect to me but perhaps just soften the wording and replace "tied".

Changed "tied" to "linked"

58 – some further clarification would be useful here on the "need to develop" – the logic isn't particularly well developed and it may be worth referencing Vongraven, D. et al. 2012. A circumpolar monitoring framework for polar bears. Ursus Monograph 5:1-66.

Thank you for the reference about a circumpolar monitoring framework for polar bears. At the end of the sentence in question, we added "e.g. as in Vongraven et al. (2012)". We think this provides sufficient justification for the "need to develop" without going into further detail about the need for monitoring polar bear habitat.

Further, it would be useful to put this study in the context of other marine species. Do the proposed metrics work for seals, seabirds, whales etc.? As written, the work is narrowly focused on polar bears yet the applicability is broader and there would be great uptake and use of the work if it could be generalized a bit.

Yes, that is a good point, thank you.  We have added a new subsection in the Discussion section:

**5.2 Relevance to other Arctic marine mammal species**

While the metrics reported here were tailored specifically to polar bears and polar bear ecology, they can be considered relevant for a range of other Arctic marine mammal (AMM) species. Besides the polar bear, AMMs are typically considered to be three cetacean species (the narwhal, *Monodon monoceros*; beluga, *Delphinapterus leucas*; and bowhead whale, *Balaena mysticetus*) and seven pinniped species (the ringed seal, *Pusa hispida*; bearded seal, *Erignathus barbatus*; spotted seal, *Phoca largha*; ribbon seal, *Phoca fasciata*; harp seal, *Pagophilus groenlandicus*; hooded seal, *Cystophora cristata*; and walrus, *Odobenus rosmarus*) (Laidre et al., 2008; Laidre et al., 2015a).  These species all occur north of the Arctic Circle for most of the year and depend on the Arctic marine ecosystem for all aspects of life.  In a few cases some may live outside the Arctic for part of the year.  All depend on the timing of sea-ice advance and retreat for different aspects of their life history, and thus the metrics in this study may be relevant to understanding changes in the regions where these AMMs occur.

We have re-numbered the other subsections in Section 5 appropriately.

67 – use the primary literature Wiig et al. 2015 is a secondary citation.

We have changed "(Wiig et al., 2015)" to "(Durner et al., 2009)" at this location in the text.

72 – "behaviorally tied" – given that the journal's audience may not be familiar with the species, some details would be useful.

The original sentence is:
"...other metrics of sea-ice habitat are more relevant to marine mammals that are behaviorally tied to the annual retreat of sea ice in the spring and advance in the fall."

We changed it to:
"...other metrics of sea-ice habitat are more relevant to marine mammals whose life history events, such as hunting and breeding, depend on the annual retreat of sea ice in the spring and advance in the fall."

75-6 – provide citations – this is a well documented area and not an original or new idea

We have added "(Stirling et al., 1999; Stirling and Parkinson, 2006)" at the end of the sentence.

89-92 – SSM/I is well documented to perform poorly during break-up and freeze-up – this issue should be discussed more fully: in particular, bias in the data. I agree, however, that the bias is likely consistent across space and time so the trends are likely valid and thus, this is not a central concern for the manuscript's validity.

Passive microwave sea-ice concentrations are well known to be biased too low over thin ice and in areas of low ice concentration, which of course are present during break-up and freeze-up. We have changed the paragraph to read as follows. Text in blue is original, text in red is new:

Concerning the accuracy of the sea-ice concentration data, the product documentation states that it is within ±5% of the actual sea-ice concentration in winter, and ±15% in summer when melt ponds are present on the sea ice; and that the accuracy is best for thick ice (> 20 cm) and high ice concentration (NSIDC, 2015). This means that accuracy is less in the marginal ice zone – the band of low ice concentration between open water and consolidated pack ice. Ivanova et al. (2015) found that all passive microwave sea-ice retrieval algorithms underestimated sea-ice concentration in the presence of melt ponds and thin ice. Thus our estimates of daily sea-ice area in each region are undoubtedly biased low, but a consistent bias over time would not affect trends computed from the data.

The final sentence of the original paragraph has been deleted, which was:
"We note that averaging over many grid cells, as is done here for the 19 regions, greatly reduces the random component of the error, although it would not reduce a bias, if present."

100-106 – the reasons for the bathymetry data are not clearly articulated. Why is bathymetry data useful? It is only clear further down (line 115) but for flow, please introduce this in the objectives.

The third paragraph of the Introduction (original lines 64-67) says:
"We calculated each metric for the total marine area of each region, and for the shallow depths only (≤ 300 m).  Shallow depths are more biologically productive and are considered to be better polar bear habitat (Durner et al., 2009)."

We think those sentences explain why bathymetry data is needed.  But we have also changed the opening sentence of the paragraph in question (original line 100) from
"For bathymetry we used ETOPO1" to
"To identify shallow depths (≤ 300 m) we used bathymetry from ETOPO1"

102 – sea ice is defined using 15% ice cover but many studies show that polar bears don't use ice cover < 30% or even 50%. Some justification from the literature is warranted.  Further, ice cover during freeze-up is different that ice use during break up (Cherry, S.G. et al. 2013. Migration phenology and seasonal fidelity of an Arctic marine predator in relation to sea ice dynamics. Journal of Animal Ecology 82:912-921.)

This comment refers to line 112, not 102.

Our use of 15% is not a statement about polar bear habitat.  We are merely calculating the area of sea ice in a region.  The 15% threshold is standard in the sea-ice literature – see, for example, Parkinson (2014).  To ignore grid cells with less than 30% or 50% sea ice would be to greatly underestimate the area of sea ice in a region.

We have added a sentence at the end of the paragraph, as follows.  Text in blue is original, text in red is new:

Sea-ice area is defined as *sea-ice concentration × grid cell area* summed over cells with sea-ice concentration greater than 15%.  For each region, we calculated the daily (or every-other-day prior to 1987) sea-ice area over two sets of grid cells: (1) all cells in the region, and (2) those cells in which the mean ocean depth is ≤ 300 meters.  We note that the 15% threshold is standard in the sea-ice literature for identifying the presence of sea ice (e.g., Parkinson, 2014) and is not based on ice concentration preferences of polar bears, which can be higher or lower depending on season (Cherry et al., 2013).

121 – what the outlier discovery was interesting, it's not central to the study and the last sentences of this paragraph could be removed.

We have condensed the last two sentences of the paragraph into one shorter sentence.

Original:
This procedure also led to the identification of an anomaly on 14 September 1984 that we reported to NSIDC, and which turned out to be an error in the passive microwave source data (see Product History at http://nsidc.org/data/docs/noaa/g02135_seaice_index/). NSIDC subsequently re-processed the data for that day.

New:
This procedure also led to the identification of an anomaly on 14 September 1984 that turned out to be an error in the passive microwave source data, which was subsequently re-processed by NSIDC.

141 – The manuscript presents data for Baffin Bay yet no reason for this was given. The other populations are available in the appendix but the focus on Baffin Bay might warrant a bit more information about this population for background. This is not essential but for those unfamiliar with the polar bears there, it may be useful context (e.g., is this an area that is ice-free is summer or not).

There is nothing special about Baffin Bay in this analysis.
We have added this parenthetical note at the end of the paragraph:
"(Figs. 2-9 use Baffin Bay as a sample region for purposes of illustration)."

Figure 2 clearly shows that Baffin Bay is ice-free in summer.

183 – "Most of the trend are statistically significant." – could you give a % (x /76) (i.e., 4 metrics for 19 populations =76) OR for each of the metrics individually (preferred).

We think it's easy enough for the reader to glance at Table 3 and see which trends are significant. They're clearly marked with * or **.

We have changed the sentence "Most of the trends are statistically significant." to "Nearly all the trends (88 of 95) are statistically significant."

183-87 – some clarification of + and – mean might be useful. The – for retreat means earlier and advance means later but "advance" being a positive number made be stop and puzzle the result. You could simply put "(i.e., earlier)" and "(i.e., later)" to aid clarity.

We have added "(negative being earlier)" and "(positive being later)"

223-229 – this is not a result but discussion. There are many sources that can be cited for this paragraph. This is not a new finding and presenting it as such ignores the literature.

We have deleted this paragraph from section 4.2 (Results) and put most of its content into section 5.4 (Discussion).

We have added this sentence at the end of the first paragraph of section 4.2:
"The negative correlations are likely the result of the ice-albedo feedback, discussed in section 5.4."

Section 5.4 now reads as follows.  Text in blue is original, text in green has been moved from section 4.2, and text in red is new.

The negative correlations between the de-trended dates of sea-ice retreat and advance (Tables 3 and 4) are likely the result of the ice-albedo feedback, noted also by Stammerjohn et al. (2012). When sea ice retreats earlier than average in spring, the ocean has more time to absorb heat from the sun.  The extra heat is stored in the upper ocean through the summer, and must be released to the atmosphere in the fall before sea ice can begin to form, thus delaying fall freeze-up. Conversely, a late spring sea-ice retreat prevents the ocean from absorbing as much heat, allowing sea ice to form earlier in the fall (e.g., Perovich et al., 2007).  The negative correlations are not perfect because other factors contribute to the timing of sea-ice retreat and advance, such as short-term weather events and long-term climate patterns.  This is also discussed in more detail by Blanchard et al. (2011), who attributed the "re-emergence of memory" in the fall to the several-month persistence of sea surface temperatures (SSTs) over the summer, enhanced by the ice-albedo feedback.  We calculated the correlation of the date of fall sea-ice advance in year $n$ with the date of spring sea-ice retreat in year $n+1$, but the correlation was not significant in any region, suggesting that SST anomalies do not persist through the winter.

285 – the present study is very similar in form and analysis to Parkinson, C.L. 2014. Spatially mapped reductions in the length of the Arctic sea ice season. Geophys Res Lett 41:4316-4322. doi:10.1002/2014GL060434. I suggest a more thorough comparison of the present manuscript and the published work is undertaken at the start of the discussion. This sort of analysis has been conducted for many polar bear population and greatly predates The Laidre et al. and Heide-Jorgensen et al. works given priority in the discussion. A fuller coverage of the issue would be useful (i.e., move the cursory treatment of similar studies forward in the discussion for proper context). This field of study has been considered for many Arctic species well before 2012 and it is appropriate to acknowledge scientific priority at the start of the discussion (i.e., not half way through in passing).

This comment refers to section 5.1, "Previous studies of the timing of Arctic sea-ice advance and retreat."  The original section consisted of seven paragraphs: 1-2. Our previous work; 3. Stammerjohn et al. (2012); 4. Parkinson (2014); 5. Frey et al. (2015); 6. Steele et al. (2015); 7. The studies listed in Table 2.  We agree that the order of the paragraphs should be changed so that scientific priority is given to previous studies of sea ice and polar bears, as listed in Table 2.

In the revised section 5.1:

-- The new paragraph #1 is now the original paragraph #7, which is about sea-ice metrics and polar bear habitat, as summarized by the 15 studies in Table 2.  We cannot discuss all 15 studies in detail, but we believe that Table 2 gives the relevant information needed to put these studies in context.  They are listed in chronological order and include the subpopulation studied, the sea-ice data source, the years of the study, the sea-ice metrics calculated, and the reference.

-- The original paragraph #7 ended with a list of seven references (all found in Table 2) that were specifically about the timing of sea-ice advance and retreat in relation to polar bears. We have added a new sentence at the end of this paragraph (now paragraph #1) as follows: "These studies are summarized in Table 2, along with eight other studies where sea-ice metrics were used for analysis of polar bear habitat."

-- The order of the paragraphs now continues as follows: 2. Stammerjohn et al. (2012); 3. Parkinson (2014); 4. Frey et al. (2015); 5. Steele et al. (2015); 6-7. Our previous work. The text in paragraphs 2-7 remains the same as in the original manuscript.

-- We agree that Parkinson (2014) is an important study, which is why we have given it a paragraph. Section 5.1 is already the longest section of the paper, so we have chosen not to include more detail.

-- Regarding "self-citation", we have moved the paragraphs about our previous work to the end of the section, so as not to imply that they represent the earliest contributions to the field.

289 – narwhal is missing its binomial name.

The binomial name now appears in the new section 5.2, "Relevance to other Arctic marine mammal species". See our response above to the comment of line 58.

344-46 – What is mean by "low variability" or "high year-to-year variability" – was this measured (could CV be used or other method (SE, SD)).

This refers to the difference between the actual dates of sea-ice advance (or retreat) and the trend line. Low variability means the dates lie close to the trend line. High variability means the dates vary more widely about the trend line. This was also stated at the beginning of section 4.2 (original lines 216-217):
"Figure 4 shows that there is year-to-year variability about the trend lines in the dates of spring sea-ice retreat and fall sea-ice advance. Subtracting out the trend lines leaves residuals."

The original wording of lines 344-346 is:
"Some regions such as East Greenland (EG) have high year-to-year variability (with respect to the trend line) in the dates of sea-ice advance and retreat, while other regions such as Foxe Basin (FB) have low variability. The high variability..."

We have changed this as follows. Original text is in blue, new text is in red:
"The dates of sea-ice advance and retreat, as shown in Figure 4 and Supplement B, vary about the trend lines. Some regions such as East Greenland (EG) have high year-to-year variability, while other regions such as Foxe Basin (FB) have low year-to-year variability (as measured, for example, by the standard deviation of the residuals about the trend line). The high variability..."

397 – "relevance to marine mammals" – non-mammalogists may not have much insight to what species you are referring to. A brief list or examples might be useful (perhaps highlighting the most sensitive species and referencing other studies for this sensitivity).

This has been addressed by the new section 5.2 – see our response to the comment of line 58 above. By the time readers reaches the "relevance to marine mammals" phrase in section 5.7, they will have already read the new section 5.2, so this phrase will make more sense.

406-420 – I found this repetitive and could be removed without loss.

This comment refers to the first two paragraphs of the Conclusions. We agree that most of it repeats what has already been presented. However, some readers will skip straight to the Conclusions and read only this section, so we believe that some repetition is OK. We have condensed these two paragraphs into one paragraph and deleted some phrases, going from 15 lines to 10 lines, which now read as follows (original text in blue, new text in red):

It is well established that the area of Arctic sea ice is declining in all months of the year, based on satellite passive microwave data from 1979 to the present (Sea Ice Index, 2016; IPCC, 2013). In this study we looked instead at the timing of sea-ice retreat in spring and advance in fall, because the duration of the sea-ice season (or equivalently the ice-free season) is important for polar bears. We found that there has been a consistent and large loss of habitat for polar bears across the Arctic. In 17 of the 19 subpopulation regions there are significant trends toward earlier spring sea-ice retreat, mostly ranging from −3 to −9 days decade$^{-1}$. In 16 of the regions there are significant trends toward later fall sea-ice advance, mostly ranging from +3 to +9 days decade$^{-1}$. Over the 3½ decades of this study, the time interval from the date of spring retreat to the date of fall advance has lengthened by 3 to 9 weeks in most regions.

430 Obbard et al. 2010 is gray literature – please use core peer-reviewed works to support this statement (of which there are many).

We have replaced "(Obbard et al., 2010)" with "(Stirling and Derocher, 2012; USFWS, 2013)" and added them to the References section:

Stirling, I. and Derocher, A. E.: Effects of climate warming on polar bears: a review of the evidence, Global Change Biology, 18: 2694–2706. doi:10.1111/j.1365-2486.2012.02753.x, 2012.

U.S. Fish and Wildlife Service (USFWS): Endangered and Threatened Wildlife and Plants; Special Rule for the Polar Bear Under Section 4(d) of the Endangered Species Act. Federal Register 78, No. 34:11766–11788, Washington, D.C., USA, 2013.

==========

**Interactive comment on "Sea-Ice Indicators of Polar Bear Habitat"**
**by H. L. Stern and K. L. Laidre**
Anonymous Referee #2

This paper was very well written and easy to follow. It is solid work with sound analyses based on the best available data, and the results are striking and present material of high value that will be important as reference material and also as a basis for future work. I have no major questions or ideas for improvement of this article. Below a couple of minor comments.

Thank you.

Introduction, line 46: "The global population size is estimated to be about 25,000..)." I am not sure if "estimated" is the best wording here, as 25,000 is the sum of several estimates and guesses/extrapolations from areas with little or no data, so maybe rewrite.

We have replaced "estimated" with "roughly estimated"

Line 47: "Genetic analysis indicates that there is considerable gene flow between some subpopulations, while others are relatively discrete (...)." I would say "shows" rather than "indicates". Genetic structure is very low. Would also rewrite last part of sentence, maybe "... It is more modest for some". I do not think subpopulations anywhere is close to being discrete.

We have re-written this sentence. The original was:
"Genetic analysis indicates that there is considerable gene flow between some subpopulations, while others are relatively discrete (Paetkau et al., 1999; Peacock et al., 2015)."

The new sentence is:
"Genetic analysis shows that gene flow occurs among the various subpopulations, which are considered to be semi-discrete (Paetkau et al., 1999; Peacock et al., 2015; Wiig et al., 2015)."

The new sentence uses "shows" rather than "indicates", and the term "semi-discrete" comes directly from Peacock et al. (2015).

Methods, page 6, line 164: "... The winter sea-ice cover will likely continue to provide suitable polar bear habitat for at least several more decades (...)." Where? In most places?

At least in the Canadian high Arctic, if not other places. We have changed the sentence as follows.

Original:
"...the winter sea-ice cover will likely continue to provide suitable polar bear habitat for at least several more decades (Wiig et al., 2015), whereas the summer sea-ice cover may not."

New:
"...the winter sea-ice cover will likely continue to provide suitable polar bear habitat for at least several more decades (especially in the Canadian high Arctic; Amstrup et al., 2008; Hamilton et al., 2014), whereas the summer sea-ice cover may not."

Discussion: page 11, line 318, "... that show an increase in the number of ice-covered days" Should be decrease, not increase? Or ice-free days, not ice-covered days?

Right, thank you!  We changed it to "a decrease in the number of ice-covered days"

==========

Other changes to the manuscript

Abstract, second-to-last sentence, we changed:
"These sea-ice metrics (or indicators of change in marine mammal habitat) were designed to be useful for management agencies."
to:
"These sea-ice metrics (or indicators of habitat change) were designed to be useful for management agencies and for comparative purposes among subpopulations."

Section 4.3, first sentence, we changed:
"The spatial pattern of trends in the date of spring sea-ice retreat (Fig. 10) shows that all trends over shallow depths are statistically significant except in the eastern and southeastern Beaufort Sea (in agreement with Steele et al., 2015) and in the north-central Canadian Arctic Archipelago."
to:
"The spatial pattern of trends in the date of spring sea-ice retreat (Fig. 10) shows that all trends over shallow depths are statistically significant except in the Northern Beaufort, Viscount Melville, and Norwegian Bay regions."

Conclusions, second paragraph, we changed "Global Climate Models (GCMs)" to "General Circulation Models (GCMs)".

In the Acknowledgements, we have added these sentence at the end:
"We thank the PBSG for input during the development of the metrics.  We thank Andy Derocher and one anonymous reviewer for comments that helped to improve the manuscript."